# Impact of Menopausal Status and Recurrent UTIs on Symptoms, Severity, and Daily Life: Findings from an Online Survey of Women Reporting a Recent UTI

**DOI:** 10.3390/antibiotics12071150

**Published:** 2023-07-05

**Authors:** Leigh N. Sanyaolu, Emily Cooper, Brieze Read, Haroon Ahmed, Donna M. Lecky

**Affiliations:** 1Division of Population Medicine, School of Medicine, Cardiff University, Cardiff CF14 4YS, UK; ahmedh2@cardiff.ac.uk; 2Primary Care and Interventions Unit, UK Health Security Agency, Twyver House, Gloucester GL1 1DQ, UK; emily.cooper@ukhsa.gov.uk (E.C.); brieze.read@ukhsa.gov.uk (B.R.); donna.lecky@ukhsa.gov.uk (D.M.L.); 3School of Medicine, Cardiff University, Cardiff CF14 4XN, UK

**Keywords:** urinary tract infection, menopause, recurrent UTIs, symptoms

## Abstract

**Introduction**: Current UKHSA UTI diagnostic guidance advises empirical antibiotics if two of the following symptoms are present: cloudy urine, dysuria, and new onset nocturia. Hormonal changes during menopause may impact UTI symptoms, and qualitative studies suggest women with recurrent UTIs may present with different UTI symptoms. This study aims to assess whether menopausal status and the presence of recurrent UTIs impact UTI symptoms in women. **Methods**: An e-survey was conducted between 13 March 2021 and 13 April 2021. Women aged 16 years or older with a history of a UTI in the last year were eligible for inclusion. We defined menopause as those aged 45–64 years; pre-menopause as those less than 45 years; and post-menopause as those 65 years and older. Recurrent UTIs were defined as three or more UTIs in the last year. The data were weighted to be representative of the UK population. Crude unadjusted and adjusted odds ratios were estimated using logistic regression. **Results**: In total, 1096 women reported a UTI in the last year. There were significant differences in UTI symptoms based on menopausal status and the presence of recurrent UTIs. Post-menopausal women self-reported more incontinence (OR 2.76, 95% CI 1.50,5.09), whereas menopausal women reported more nocturia. Women with recurrent UTIs reported less dysuria, more severe symptoms (OR 1.93 95% CI 1.37,2.73) and a greater impact on daily life (OR 1.68, 95% CI 1.19,2.37). **Conclusions**: This survey provides evidence that acute UTIs present differently based on menopausal status and in women with recurrent UTIs. It is important that healthcare professionals are aware of these differences when assessing women presenting with an acute UTI and, therefore, further research in this area is needed.

## 1. Introduction

Urinary tract infections (UTIs) in women have an estimated annual incidence of 10–13% with more than 50% of women experiencing at least one UTI during their lifetime [1,2,3]. Around 95% of women with a UTI see a healthcare professional, accounting for 1–3% of all UK primary care consultations [2,4,5]. The primary care diagnosis and management of UTIs is often based on symptoms alone or in conjunction with urine testing [6,7]. Current UK Health Security Agency (UKHSA) evidence-based guidance advises empirical antibiotics if two or more of the symptoms of cloudy urine, dysuria, and new onset nocturia are present [7,8]. There are concerns about empirical antibiotic use, especially since evidence suggests a significant proportion of antibiotics are prescribed inappropriately [9,10]. This has serious ramifications since antibiotic consumption is a major driver of antimicrobial resistance (AMR) [11]. Accurate diagnosis of acute infection is, therefore, a crucial step towards more appropriate antibiotic prescribing and reduced burden of AMR.

It is known that symptoms of an acute UTI may differ in those with cognitive impairment but not whether they differ in other patient populations. Evidence shows that UTI symptoms and their diagnostic properties can change with age [6,12]. The number of UTIs in women over 65 years increases over time, and the incidence of UTIs increases around menopause [13,14]. Menopause is a time of significant hormonal change and can result in the genitourinary syndrome of menopause (GSM) [15,16,17,18]. The symptoms of GSM include vaginal dryness and urinary symptoms, such as urgency, dysuria, and frequency [15,16,17,18]. GSM is seen in up to 50% of post-menopausal and is often chronic and progressive, starting 4–5 years after menopause [16,17,18]. Combined with the increasing prevalence of asymptomatic bacteriuria in older women, leading to false positive results, this renders urine sampling less effective and highlights the need for a greater understanding of UTI symptoms in this population [19,20].

Recurrent UTIs, defined as two UTIs in six months or three in twelve months, occur in up to 3% of women [2,21,22,23]. Women with recurrent UTIs are often prescribed repeated courses of antibiotics and, as such, have a greater risk of AMR [24,25,26,27]. Research into differences in the acute UTI symptoms of patients with recurrent UTIs is lacking and is limited to qualitative studies suggesting both ‘typical’ and ‘atypical’ symptoms [25,27]. It is, therefore, important to understand if the symptoms differ in women with recurrent UTIs to ensure the appropriate antibiotic treatment.

This study aims to assess whether menopausal status and the presence of recurrent UTIs impact the presenting symptoms of women with self-reported UTIs using an online survey.

## 2. Materials and Methods

### 2.1. Study Design

This was an online survey (e-survey) of adult women. The survey was based on a previously delivered computer-assisted in-person survey (2014) and was adapted to include information relevant to the COVID-19 pandemic [2].

### 2.2. Participants and Setting

The survey was administered between 13 March 2021 and 13 April 2021. The participants were recruited via Ipsos’ online panels. Ipsos (https://www.ipsos.com/en-uk, accessed on 30 June 2023) is a global market research group that utilises a variety of research methods, including in-person and telephone interviewing and internet surveys. For this survey, Ipsos utilised multi-source recruitment methods [28,29]. Eligible participants were defined as women aged 16 years or older with a previous experience of a UTI in the last 12 months. The definition of a UTI was clarified at the outset with the following statement: *Urinary Tract Infections (UTIs) are often called urine, water, or bladder infections or cystitis*. *They can give you pain when passing urine and a need to pass urine more often.*

### 2.3. Exposures and Outcomes

We assessed the associations between menopausal status and recurrent UTIs (exposures) and self-reported symptoms, symptom severity, and impact on daily life (outcomes).

As menopause typically occurs between the ages of 45 and 55 years (mean 55 years) and GSM occurs 4–6 years after the onset of menopause, we categorised women aged 45–64 as menopausal. This is comparable with other research [12,30]. We defined women aged less than 45 years as pre-menopausal and those aged 65 years and older as post-menopausal. Recurrent UTIs were defined as three or more UTIs reported in the last year.

The outcomes were related to the participants’ most recent UTI and the participants could respond with multiple symptoms associated with this UTI. The symptom severity was assessed by asking the participants “*Thinking about your most recent UTI, how severe, if at all, were your symptoms? Please give your answer on a scale of 0 to 10 where 0 means you had no symptoms at all and 10 means your symptoms were very severe*”. The severity was recorded on a 0 to 10 Likert scale. The impact on daily life was assessed by asking the participants “*To what extent, if at all, did your most recent UTI affect your daily life?*”. The possible responses included: it affected it a great deal, it affected it a fair amount, it didn’t affect it very much, or it didn’t affect it at all.

### 2.4. Analysis

The data were weighted by age, ethnicity, working status, housing tenure, social grade, education, and Government Office Region (GOR) to be representative of the UK population. The initial analyses were conducted by Ipsos. Additional analyses were performed using STATA v 14 (EC). Logistic regression models estimated the crude and adjusted odds ratios (OR) and 95% confidence intervals (CI) for each self-reported symptom and the presence of two or more of the three UTI predictive symptoms (cloudy urine, dysuria, and new onset nocturia) based on menopausal and recurrent UTI status. Ordinal logistic regression was used to assess how menopausal and recurrent UTI status influenced the symptom severity and quality of life. The adjusted ORs accounted for the age group, social grade, region, marital status, education, employment status, number of children, and ethnicity. Forest plots were created using R version 4.3.1.

## 3. Results

Approximately 53,000 women aged 16 years and over were approached to complete the survey. A total of 4153 women participated in the online survey with 2542 (61%) stating they had previously had a UTI, and of these, 1096 (26%) reported having a UTI in the last year and formed the study population. The mean age was 45 years, 90% (n = 988) were White, 30% (n = 327) were from North England, 30% (n = 327) were from the Midlands, 63% (n = 690) were married, and 636 (58%) were employed (Table 1).

The most common self-reported symptoms were dysuria (65%, n = 713) and urinary frequency (65%, n = 715), followed by lower abdominal pain (50%, n = 549), nocturia (47%, n = 515), and cloudy urine (37%, n = 407) (Figure 1).

### 3.1. Menopausal Status and Self-Reported Symptoms

Compared to pre-menopausal women (reference group), menopausal women were more likely to report urinary frequency (OR 1.44, 95% confidence interval (CI) 1.03,2.00), nocturia (OR 1.43, 95% CI 1.05,1.96), or new or worsening urinary incontinence (OR 1.66, 95% CI 1.06,2.62) (Figure 2). Post-menopausal women were more likely to report new or worsening urinary incontinence (OR 2.76, 95% CI 1.50,5.09) (Figure 3). Menopausal women were less likely to report haematuria (OR 0.58, 95% CI 0.36,0.94), vaginal discharge (OR 0.25, 95% CI 0.16,0.40), and unsteadiness (OR 0.55, 95% CI 0.31,0.99). Post-menopausal women were less likely to report dysuria (OR 0.61, 95% CI 0.38,0.97), lower abdominal pain (OR 0.56, 95% CI 0.35,0.87), haematuria (OR 0.46, 95% CI 0.23,0.93), vaginal discharge (OR 0.15, 95% CI 0.07,0.31), fever (OR 0.25, 95% CI 0.11,0.60), or kidney pain (OR 0.58, 95% CI 0.34,0.99). There was no difference by menopausal status for the presence of two or more of the strongly predictive symptoms of UTIs (dysuria, nocturia, or cloudy urine). Menopausal women were more likely to report greater severity of symptoms compared to pre-menopausal women (OR 1.39, 95% CI 1.04,1.86), whereas post-menopausal women reported a lesser impact on daily life compared to pre-menopausal symptoms (OR 0.62, 95% CI 0.39,0.98) (Table 2 and Table 3).

### 3.2. Recurrent UTIs and Self-Reported Symptoms

Compared to women with 1–2 UTIs in the previous year, women with recurrent UTIs were less likely to report dysuria (OR 0.63, 95% CI 0.44,0.89) (Figure 4) but were more likely to report unsteadiness (OR 1.85, 95% CI 1.10,3.10), shivering, shaking or rigours (OR 2.30, 95% CI 1.42,3.73), and kidney pain (OR 1.71, 95% CI 1.18,2.47). There was no difference between women with and without recurrent UTIs and the reporting of the strongly predictive symptoms of UTIs. Women with recurrent UTIs were more likely to report greater severity (OR 1.93 95% CI 1.37,2.73) (Table 2) and a greater impact on their daily life (OR 1.68, 95% CI 1.19,2.37) (Table 3).

## 4. Discussion

This online survey of 1096 women found differences in the acute UTI symptoms experienced by women based on menopausal status or having recurrent UTIs. We found that compared to pre-menopausal women, post-menopausal women were more likely to self-report new or worsening incontinence and less likely to report fever and abdominal pain. Menopausal women were more likely to report new nocturia and incontinence. Women with recurrent UTIs were less likely to report dysuria but more likely to report unsteadiness, shivering, shaking or rigours, and kidney pain. There was no difference based on menopausal or recurrent UTIs for reporting two or more of the three strongly predictive symptoms of a UTI. Women with recurrent UTIs self-reported more severe symptoms of their UTIs and a greater impact on life compared to those without recurrent UTIs. We found possible associations for other symptoms and the severity and impact on daily life based on menopausal status. However, we have less confidence in these findings as the OR confidence intervals were close to 1.

### 4.1. Comparison with Existing Literature

Several studies have evaluated the use of specific symptoms to diagnose UTIs [8,31,32]. Although various symptoms have been shown to be predictive of UTIs, symptom combinations are more predictive [8,31,32]. UKHSA guidance advises empirical antibiotics if two or more of the symptoms of cloudy urine, dysuria, and new onset nocturia are present, based on research that revealed that the positive predictive value of two or more symptoms was 74% and three or more symptoms was 82% [7,8]. However, it is unclear if the predictive value of these symptoms differs based on menopausal status or the presence of recurrent UTIs.

The results of our study provide some evidence that the symptoms of an acute UTI differ by menopausal status. Few studies have investigated this in detail. An observational study investigated 196 community-dwelling women in Israel between 2009 and 2010 and compared the acute UTI symptoms of women aged 45–54 years (defined as pre-menopausal) to those of women over 65 years (defined as post-menopausal) [12]. They found that post-menopausal women were more likely to report “urgency of urination”, “pain with voiding”, “incontinence”, “lower abdominal”, and “low back pain” but less likely to report “urinary frequency” and “painful” or “burning urination”. Our study demonstrates similarities in terms of incontinence and dysuria but the opposite finding in terms of abdominal pain. They also reported that post-menopausal women, compared to pre-menopausal women, reported more “frequency” and “urgency”, “back pain”, and “cold chills”. We found no increasing odds for such symptoms. Our study did, however, find that women of menopausal age reported more frequency, nocturia, or incontinence. Possible explanations for this discrepancy could be how “bladder pain” and “abdominal pain” were defined and how “painful urination”, “burning urination”, and “painful voiding” were distinguished, as this is not clear. Additionally, how pre-menopause was defined also differed. The findings of menopausal women being more likely to report new onset nocturia could be explained by the onset of menopause and GSM [15,16,17,18].

Another observational study of 1178 women presenting to Danish General Practice with UTI symptoms found that the symptoms were similar between age groups, although the reporting of dysuria decreased with age [6]. A questionnaire-based study of 661 women from UK General Practice similarly found that older women were less likely to report dysuria compared to younger women [33]. Both studies looked at age rather than menopausal status but appear to show less reported dysuria with increasing age. We found no clear association between menopausal status and dysuria. The reason why the presence of dysuria would change with age or menopause is not clear. Some potential explanations include reduced immune system function with increasing age, age and menopausal-related changes to the urinary tract, and the presence of asymptomatic bacteriuria [34].

The finding of post-menopausal women being less likely to self-report fever was not reported in the studies mentioned above but may be related to age rather than menopausal status. There is evidence that in the older population, between 20 and 30% will not have a fever with acute infection and that it has a limited diagnostic value for UTIs in the elderly [34,35,36,37]. The reasons for this blunted temperature response are not clear but may partially explain the results seen in our survey.

To date, no study has evaluated if women with recurrent UTIs present differently from women without recurrent UTIs. Evidence from qualitative research suggests that women with recurrent UTIs describe the typical symptoms of UTIs, such as dysuria, but also describe more atypical and generalised symptoms associated with UTIs, including shivering and weakness [25,27]. Our results show that women with recurrent UTIs report less dysuria (the reasons for which are unclear) whilst also reporting more non-specific symptoms associated with UTIs, like shivering, aligning with the qualitative evidence [25]. We also found that women with recurrent UTIs reported their symptoms as more severe compared to women without recurrent UTIs, and in the qualitative studies, women with recurrent UTIs often describe their UTIs as severe with a significant impact on their daily life [25,27].

### 4.2. Strengths and Limitations

This online survey study has several strengths and limitations. The surveyed population size was large and included 4153 participants from across England. It had a good representation of all age groups, social classes, and ethnic groups. Being an online survey may reduce the social desirability bias compared to face-to-face methods. Nearly two-thirds of women in this study (63%, n = 686) received antibiotics for a suspected UTI, but we aimed to explore the symptom experiences in women with a recent self-diagnosed UTI. Therefore, we did not restrict the participants to those who had a clinically diagnosed UTI, and we believe this is a strength of this study as it allowed us to assess women who chose not to seek care for their symptoms. There is a risk of recall bias with our survey since it relates to the respondents’ last UTI; however, this is reduced by focusing only on those with a UTI in the last year. We searched for any association between the severity score and the time of the participant’s last UTI and found that the severity scores were broadly similar, irrespective of the time from the last UTI (Appendix A). The severity of the UTI symptoms was assessed using a 10-point Likert scale that was not validated and so might not accurately reflect the severity of the symptoms in this population. Finally, there is a risk of selection bias since certain populations may have been excluded due to limited internet access, the inability to complete the questionnaire online, or based on their survey preferences.

### 4.3. Implications for Clinical Practice and Future Research

Dysuria and nocturia are two of the three strongly predictive symptoms; however, women with recurrent UTIs have lower odds of reporting dysuria and menopausal women have higher odds of reporting nocturia. This has implications for the diagnosis and treatment of acute UTIs in these patient populations and suggests that a one-size-fits-all approach may not be suitable, and any assessment of acute UTI symptoms should consider the menopausal status and history of recurrent UTIs. The self-reporting of two of the three predictive symptoms of an acute UTI did not appear to differ in these patient populations, providing some evidence that the current UKHSA guidance remains applicable. There is very limited research in these areas, and further exploration is required to ensure the clinical guidelines reflect the differences in UTI symptoms within these populations.

## 5. Conclusions

This online survey provides evidence that acute UTIs present differently based on menopausal status and if women have recurrent UTIs. Post-menopausal women self-reported more incontinence, whereas menopausal women reported more nocturia and women with recurrent UTIs reported less dysuria, which are two of the predictive symptoms. Healthcare professionals need to be aware of how UTI symptoms differ in these patients, and further research is required to confirm that current guidance is applicable in these populations to ensure appropriate antibiotic stewardship.

## Figures and Tables

**Figure 1 antibiotics-12-01150-f001:**
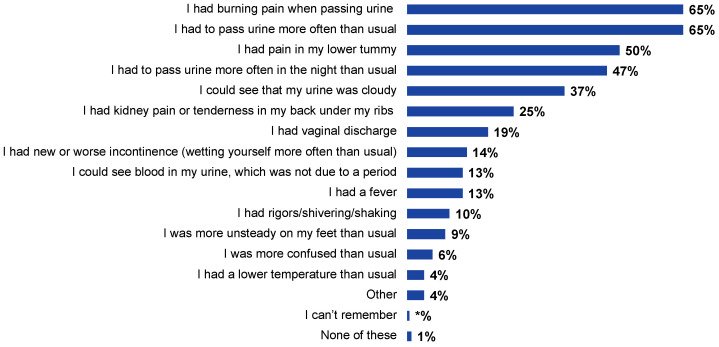
Self-reported symptoms in response to the question “Thinking about your most recent UTI, did you have any of the following signs or symptoms?”. Note that respondents could answer with multiple symptoms (n = 1096).

**Figure 2 antibiotics-12-01150-f002:**
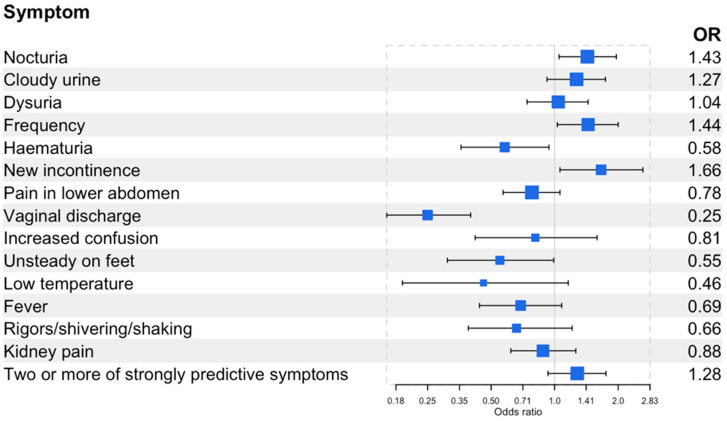
Forest plot of the odds of having the self-reported symptoms in menopausal (45–64 years, n = 332) women compared to pre-menopausal (16–44 years, n = 570) women. Two or more of the strongly predictive symptoms include cloudy urine, dysuria, and new onset nocturia. See Appendix A for the prevalence of self-reported symptoms and odds of having these symptoms.

**Figure 3 antibiotics-12-01150-f003:**
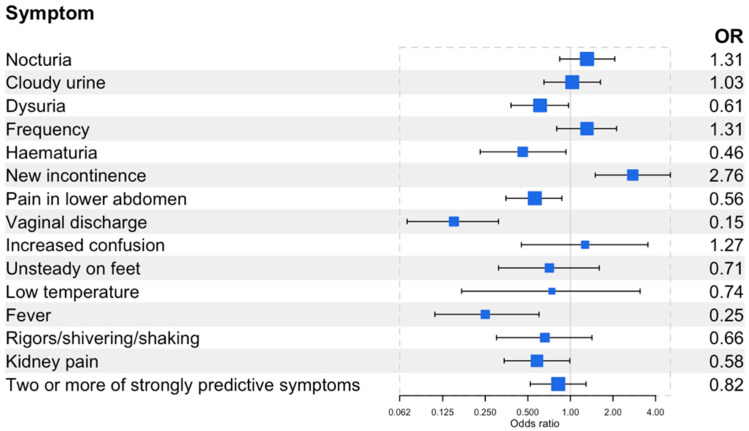
Forest plot of the odds of having the self-reported symptoms in post-menopausal women (>65 years, n = 194) compared to pre-menopausal (16–44 years, n = 570) women. Two or more of the strongly predictive symptoms include cloudy urine, dysuria, and new onset nocturia. See Appendix A for the prevalence of self-reported symptoms and odds of having these symptoms.

**Figure 4 antibiotics-12-01150-f004:**
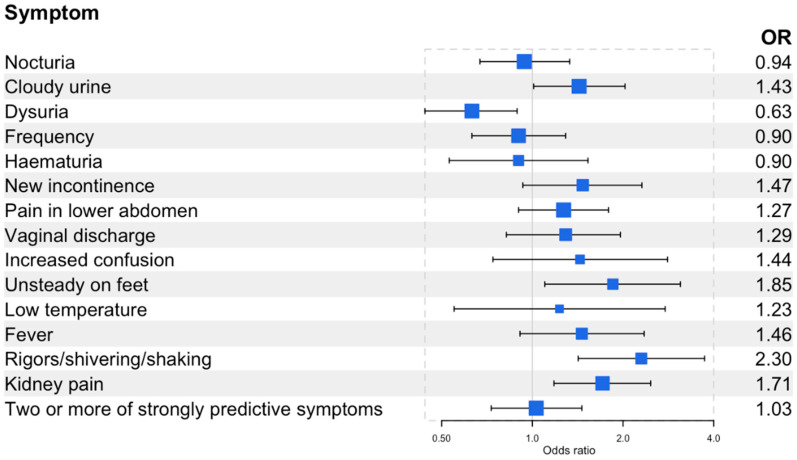
Forest plot of the odds of having the self-reported symptoms in women with recurrent UTIs (3 or more UTIs in the previous year, n = 164) compared to those without recurrent UTIs (1–2 UTIs in the previous year, n = 931). Two or more of the strongly predictive symptoms include cloudy urine, dysuria, and new onset nocturia. See Appendix A for the prevalence of self-reported symptoms and the odds of having these symptoms.

**Table 1 antibiotics-12-01150-t001:** Demographics of e-survey study population (unweighted).

Demographics		e-Survey Study Population (%) n = 1096
Age (years)	16–24	130 (12%)
	25–34	254 (23%)
	35–44	183 (17%)
	45–54	205 (19%)
	55–64	133 (12%)
	65–74	119 (11%)
	Over 75	72 (7%)
Ethnicity	White	988 (90%)
	Ethnic minorities	101 (9%)
Social grade	AB	284 (26%)
	C1	360 (33%)
	C2	159 (15%)
	DE	293 (27%)
Region	North England	327 (30%)
	Midlands	327 (30%)
	South England	287 (26%)
	London	155 (14%)
Marital status	Married/living as married	690 (63%)
	Single	253 (23%)
	Widowed/divorced/separated	153 (14%)
Education	GCSE/O-level/NVQ12	279 (25%)
	A-level or equivalent	276 (25%)
	Degree/Masters/PhD	401 (37%)
	No formal qualifications	140 (13%)
Employment status	Employed (full-time/part-time or self-employed)	636 (58%)
	Unemployed	460 (42%)
Children in the household (17 years of age or under)	No children	708 (65%)
	1 child	204 (19%)
	2 children	130 (12%)
	3 or more children	54 (5%)

**Table 2 antibiotics-12-01150-t002:** Severity of UTI symptoms according to menopausal status and presence of recurrent UTIs. Six participants responded ‘Don’t know’ to severity based on menopausal status (four in pre-menopausal group and two in the post-menopausal group). Six participants responded ‘Don’t know’ to severity based on presence of recurrent UTIs (four in the 1–2 UTIs in the previous year group and two in the 3 or more UTIs in the past year group.

		Severity Score N (%)											Adjusted OR (95% CI)
		0	1	2	3	4	5	6	7	8	9	10	
Menopausal status	Pre-menopause	5 (1%)	11 (2%)	17 (3%)	40 (7%)	45 (8%)	80 (14%)	132 (23%)	113 (20%)	71 (13%)	35 (6%)	19 (3%)	Reference
	Menopause	2 (1%)	3 (1%)	8 (2%)	21 (6%)	15 (4%)	52 (16%)	59 (18%)	79 (24%)	59 (18%)	20 (6%)	14 (4%)	1.39 (1.04, 1.86)
	Post-menopause	4 (2%)	1 (0.5%)	6 (3%)	12 (6%)	20 (11%)	25 (13%)	22 (12%)	48 (25%)	31 (16%)	11 (5%)	11 (6%)	1.20 (0.78, 1.85)
UTIs in the previous year	1–2 UTIs	7 (1%)	14 (1%)	29 (3%)	71 (8%)	70 (8%)	134 (14%)	186 (20%)	207 (22%)	134 (14%)	45 (5%)	31 (3%)	Reference
	3 or more UTIs	4 (2%)	1 (0.5%)	2 (1%)	3 (2%)	10 (6%)	23 (14%)	27 (17%)	33 (20%)	28 (17%)	20 (12%)	13 (8%)	1.93 (1.37, 2.73)

**Table 3 antibiotics-12-01150-t003:** Impact on daily life of UTI symptoms according to menopausal status and presence of recurrent UTIs. Three participants responded ‘Don’t know’ to impact on quality of life according to menopausal status (all in the pre-menopausal group). Three participants responded ‘Don’t know’ to impact on quality of life based on presence of recurrent UTIs (all in the 1–2 UTIs in the previous year group).

		Impact on Daily Life N (%)				Adjusted OR (95% CI)
		None	Not Much	Fair Amount	Great Deal	
Menopausal status	Pre-menopause	19 (3%)	125 (22%)	307 (54%)	117 (21%)	Reference
	Menopause	7 (2%)	72 (22%)	184 (55%)	69 (21%)	1.09 (0.80, 1.47)
	Post-menopause	11 (6%)	64 (33%)	86 (45%)	33 (17%)	0.62 (0.39, 0.98)
UTIs in the previous year	1–2 UTIs	31 (3%)	234 (25%)	491 (53%)	173 (19%)	Reference
	3 or more UTIs	5 (3%)	27 (16%)	86 (53%)	46 (28%)	1.68 (1.19, 2.37)

## Data Availability

All data requests should be submitted to the corresponding author for consideration. Access to the data may be granted following review.

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
