# Peer review of "Impact of Menopausal Status and Recurrent UTIs on Symptoms, Severity, and Daily Life: Findings from an Online Survey of Women Reporting a Recent UTI"

_antibiotics, 2023, doi:10.3390/antibiotics12071150_

Round 1

Reviewer 1 Report

Thank you for the opportunity to review this paper. I find it academically interesting. However, with some changes, it may also become clinically interesting and more readable.

Outcome measure: It seems you have not used a validated outcome measure. This is a clear weakness of the study and you will need to describe our outcome in more detail including how it was developed, if it was tested in cognitive interviews, if the women found a 10-point scale relevant ect. You also need to elaboaret a lot more on this under limitations.

You calculate severity scores based on the sum of all symptoms? To do this, you should have investigated whether your items have the psychometric properties allowing for sum-scoring. Otherwice it is better to report them separately.

Women had to recall their latest UTI and rate their symptoms on a 10-point likert scale? You need to report how long ago the most recent UTI was and if this had any impact on the rating of symptoms. I would imagine people have some trouble recalling the severity of each symptom 10 moths later and that these also change during the URI period. Also a point for limitations.

I do not think it is a limitation that UTI was “self-diagnosed”. You aim to investigate symptom experiences, so this is more a strength.

“Data were weighted by age, ethnicity, working status, housing tenure, social grade, 93 education, and Government Office Region (GOR) to be representative of the UK population”. The UK population who get UTIs? Otherwice this is not really relevant. Why should your sample be representative for someone who never experience UTIs?

I cant find your 10-point scale in your results. How did you dichotomize it? Did you just use if they experienced it or not? That is not stated in your methods.

Your tables are quite large and complex. Perhaps a figure would be better at showing which tendencies you have found? This is more out of courtesy to improve the reading experience, not something you have to change if you prefer the tables.

“This online survey of 1096 women found substantial differences in the acute UTI 155 symptoms experienced by women based on menopausal status or having recurrent UTIs.” How are they substantial? You find some differences here and there, which is interesting. But if it is clinically interesting depends on the difference in severity which you do not seem to have reported? You have a large survey – perhaps it is a difference between 0 and 1,2 on a 10-point scale. That is probably not very substantial. I still find your findings interesting, but at present only academically.

Reviewer 2 Report

This study presents an interesting observation about the influence of menopausal status on symptoms of UTIs and has some merit to be published. However the reviewer have some concerns regarding this study.

It is understandable that, as a practical basis, UTIs are diagnosed based on symptoms alone and treated by antibiotics. However, without any laboratory tests, the diagnosis of UTIs is not definite, and in the era of AMR, such symptom-based approach may not be sustainable any longer.

For the definite diagnosis of UTIs urine culture would be necessary. However it could be argued that performing urine culture to all UTI cases would not be cost-effective. Urine culture could be limited to cases of recurrent UTIs as was suggested by the literature (Michael L. Wilson Loretta Gaido, Laboratory Diagnosis of Urinary Tract Infections in Adult Patients, Clinical Infectious Diseases, Volume 38, Issue 8, 15 April 2004, Pages 1150–1158, https://doi.org/10.1086/383029).

In the reviewer's opinion, the diagnosis of UTIs should be made based on both both clinical symptoms and laboratory tests such as urine dip-stick tests, especially in cases of recurrent UTIs. The findings of this study emphasize the importance of laboratory tests, since the typical symptoms of UTIs are not always present in menopausal women and women with recurrent UTIs.  

The authors should discuss this point, and include relevant citations. In addition, if any information regarding laboratory test in the patient cohort is available, the data should also be analyzed with regard to the symptoms of the patients.

Round 2

Reviewer 2 Report

This study would be applicable solely in the medical practice in UK. Medical practitioners outside of UK could hardly obtain novel insights to improve their practice from this study. The reviewer advises the authors to submit this manuscript to local medical journals, not to international journal such as Antibiotics.